# Exponential suppression of bit or phase errors with cyclic error correction

Google Quantum AI*

Realizing the potential of quantum computing requires sufficiently low logical error rates[1]. Many applications call for error rates as low as $10^{-15}$ (refs. [2–9]), but state-of-the-art quantum platforms typically have physical error rates near $10^{-3}$ (refs. [10–14]). Quantum error correction[15–17] promises to bridge this divide by distributing quantum logical information across many physical qubits in such a way that errors can be detected and corrected. Errors on the encoded logical qubit state can be exponentially suppressed as the number of physical qubits grows, provided that the physical error rates are below a certain threshold and stable over the course of a computation. Here we implement one-dimensional repetition codes embedded in a two-dimensional grid of superconducting qubits that demonstrate exponential suppression of bit-flip or phase-flip errors, reducing logical error per round more than 100-fold when increasing the number of qubits from 5 to 21. Crucially, this error suppression is stable over 50 rounds of error correction. We also introduce a method for analysing error correlations with high precision, allowing us to characterize error locality while performing quantum error correction. Finally, we perform error detection with a small logical qubit using the 2D surface code on the same device[18,19] and show that the results from both one- and two-dimensional codes agree with numerical simulations that use a simple depolarizing error model. These experimental demonstrations provide a foundation for building a scalable fault-tolerant quantum computer with superconducting qubits.

Many quantum error-correction (QEC) architectures are built on stabilizer codes[20], where logical qubits are encoded in the joint state of multiple physical qubits, which we refer to as data qubits. Additional physical qubits known as measure qubits are interlaced with the data qubits and are used to periodically measure the parity of chosen data qubit combinations. These projective stabilizer measurements turn undesired perturbations of the data qubit states into discrete errors, which we track by looking for changes in parity. The stream of parity values can then be decoded to determine the most likely physical errors that occurred. For the purpose of maintaining a logical quantum memory in the codes presented in this work, these errors can be compensated in classical software[3]. In the simplest model, if the physical error per operation $p$ is below a certain threshold $p_{th}$ determined by quantum computer architecture, chosen QEC code and decoder, then the probability of logical error per round of error correction ($\varepsilon_L$) should scale as:

$$\varepsilon_L = C/\Lambda^{(d+1)/2}. \tag{1}$$

Here, $\Lambda \propto p_{th}/p$ is the exponential error suppression factor, $C$ is a fitting constant and $d$ is the code distance, defined as the minimum number of physical errors required to generate a logical error, and increases with the number of physical qubits[3,21]. More realistic error models cannot be characterized by a single error rate $p$ or a single threshold value $p_{th}$. Instead, quantum processors must be benchmarked by measuring $\Lambda$.

Many previous experiments have demonstrated the principles of stabilizer codes in various platforms such as nuclear magnetic resonance[22,23], ion traps[24–26] and superconducting qubits[19,21,27,28]. However, these results cannot be extrapolated to exponential error suppression in large systems unless non-idealities such as crosstalk are well understood. Moreover, exponential error suppression has not previously been demonstrated with cyclic stabilizer measurements, which are a key requirement for fault-tolerant computing but introduce error mechanisms such as state leakage, heating and data qubit decoherence during measurement[21,29].

In this work, we run two stabilizer codes. In the repetition code, qubits alternate between measure and data qubits in a 1D chain, and the number of qubits for a given code distance is $n_{qubits} = 2d - 1$. Each measure qubit checks the parity of its two neighbours, and all measure qubits check the same basis so that the logical qubit is protected from either $X$ or $Z$ errors, but not both. In the surface code[3,30–32], qubits follow a 2D chequerboard pattern of measure and data qubits, with $n_{qubits} = 2d^2 - 1$. The measure qubits further alternate between $X$ and $Z$ types, providing protection against both types of errors. We use repetition codes up to $d = 11$ to test for exponential error suppression and a $d = 2$ surface code to test the forward compatibility of our device with larger 2D codes.

## QEC with the Sycamore processor

We implement QEC using a Sycamore processor[33], which consists of a 2D array of transmon qubits[34] where each qubit is tunably coupled to four nearest neighbours—the connectivity required for the surface

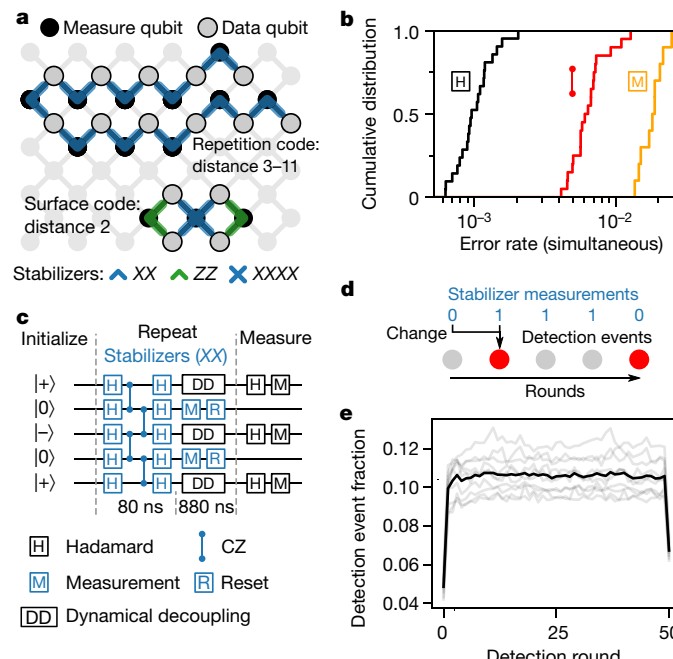

**a** ● Measure qubit ○ Data qubit

Repetition code: distance 3–11

Surface code: distance 2

Stabilizers: ▲ *XX* ▲ *ZZ* ✕ *XXXX*

**c**

| Initialize | Repeat | Measure |
|---|---|---|

Stabilizers (*XX*)

$|+\rangle$
$|0\rangle$
$|-\rangle$
$|0\rangle$
$|+\rangle$

80 ns    880 ns

Ⓗ Hadamard ▐ CZ

Ⓜ Measurement Ⓡ Reset

DD Dynamical decoupling

**b** Cumulative distribution vs Error rate (simultaneous)

**d** Stabilizer measurements
0  1  1  1  0
Change → Detection events
Rounds

**e** Detection event fraction vs Detection round

**Fig. 1 | Stabilizer circuits on Sycamore. a**, Layout of distance-11 repetition code and distance-2 surface code in the Sycamore processor. In the experiment, the two codes use overlapping sets of qubits, which are offset in the figure for clarity. **b**, Pauli error rates for single-qubit and CZ gates and identification error rates for measurement, each benchmarked in simultaneous operation. **c**, Circuit schematic for the phase-flip code. Data qubits are randomly initialized into $|+\rangle$ or $|-\rangle$, followed by repeated application of *XX* stabilizer measurements and finally *X*-basis measurements of the data qubits. Hadamard refers to the Hadamard gate, a quantum operation. **d**, Illustration of error detection events that occur when a measurement disagrees with the previous round. **e**, Fraction of measurements (out of 80,000) that detected an error versus measurement round for the $d = 11$ phase-flip code. The dark line is an average of the individual traces (grey lines) for each of the 10 measure qubits. The first (last) round also uses data qubit initialization (measurement) values to identify detection events.

code. Compared with ref. [33], this device has an improved design of the readout circuit, allowing for faster readout with less crosstalk and a factor of 2 reduction in readout error per qubit (see Supplementary Section I). Like its predecessor, this processor has 54 qubits, but we used at most 21 because only a subset of the processor was wired up. Figure 1a shows the layout of the $d = 11$ (21 qubit) repetition code and $d = 2$ (7 qubit) surface code on the Sycamore device, while Fig. 1b summarizes the error rates of the operations which make up the stabilizer circuits. Additionally, the mean coherence times for each qubit are $T_1 = 15 \,\mu s$ and $T_2 = 19 \,\mu s$.

The experiments reported here leverage two recent advancements in gate calibration on the Sycamore architecture. First, we use the reset protocol introduced in ref. [35], which removes population from excited states (including non-computational states) by sweeping the frequency of each measure qubit through that of its readout resonator. This reset operation is appended after each measurement in the QEC circuit and produces the ground state with error below 0.5%[35] in 280 ns. Second, we implement a 26-ns controlled-*Z* (CZ) gate using a direct swap between the joint states $|1, 1\rangle$ and $|0, 2\rangle$ of the two qubits (refs. [14,36]). As in ref. [33], the tunable qubit–qubit couplings allow these CZ gates to be executed with high parallelism, and up to 10 CZ gates are executed simultaneously in the repetition code. Additionally, we use the results of running QEC to calibrate phase corrections for each CZ gate (Supplementary Information section III). Using simultaneous cross-entropy

benchmarking[33], we find that the median CZ gate Pauli error is 0.62% (median CZ gate average error of 0.50%).

We focused our repetition code experiments on the phase-flip code, where data qubits occupy superposition states that are sensitive to both energy relaxation and dephasing, making it more challenging to implement and more predictive of surface code performance than the bit-flip code. A five-qubit unit of the phase-flip code circuit is shown in Fig. 1c. This circuit, which is repeated in both space (across the 1D chain) and time, maps the pairwise *X*-basis parity of the data qubits onto the two measure qubits, which are measured then reset. During measurement and reset, the data qubits are dynamically decoupled to protect the data qubits from various sources of dephasing (Supplementary Section XI). In a single run of the experiment, we initialize the data qubits into a random string of $|+\rangle$ or $|-\rangle$ on each qubit. Then, we repeat stabilizer measurements across the chain over many rounds, and finally, we measure the state of the data qubits in the *X* basis.

Our first pass at analysing the experimental data is to turn measurement outcomes into error detection events, which are changes in the measurement outcomes from the same measure qubit between adjacent rounds. We refer to each possible spacetime location of a detection event (that is, a specific measure qubit and round) as a detection node. In Fig. 1e, for each detection node in a 50-round, 21-qubit phase-flip code, we plot the fraction of experiments (80,000 total) where a detection event was observed on that node. This is the detection event fraction. We first note that the detection event fraction is reduced in the first and last rounds of detection compared with other rounds. At these two time boundary rounds, detection events are found by comparing the first (last) stabilizer measurement with data qubit initialization (measurement). Thus, the data qubits are not subject to decoherence during measure qubit readout in the time boundary rounds, illustrating the importance of running QEC for multiple rounds in order to benchmark performance accurately (Supplementary Information section VII). Aside from these boundary effects, we observe that the average detection event fraction is 11% and is stable across all 50 rounds of the experiment, a key finding for the feasibility of QEC. Previous experiments had observed detections rising with number of rounds[21], and we attribute our experiment's stability to the use of reset to remove leakage in every round[35].

## Correlations in error detection events

We next characterize the pairwise correlations between detection events. With the exception of the spatial boundaries of the code, a single-qubit Pauli error in the repetition code should produce two detections which come in three categories[21]. First, an error on a data qubit usually produces a detection on the two neighbouring measure qubits in the same round—a spacelike error. The exception is a data qubit error between the two CZ gates in each round, which produces detection events offset by 1 unit in time and space—a spacetimelike error. Finally, an error on a measure qubit will produce detections in two subsequent rounds—a timelike error. These error categories are represented in the planar graph shown in Fig. 2a, where expected detection pairs are drawn as graph edges between detection nodes.

We check how well Sycamore conforms to these expectations by computing the correlation probabilities between arbitrary pairs of detection nodes. Under the assumptions that all correlations are pairwise and that error rates are sufficiently low, we estimate the probability of simultaneously triggering two detection nodes $i$ and $j$ as

$$p_{ij} \approx \frac{\langle x_i x_j \rangle - \langle x_i \rangle \langle x_j \rangle}{(1 - 2\langle x_i \rangle)(1 - 2\langle x_j \rangle)}, \tag{2}$$

where $x_i = 1$ if there is a detection event and $x_i = 0$ otherwise, and $\langle x \rangle$ denotes an average over all experiments (Supplementary Information

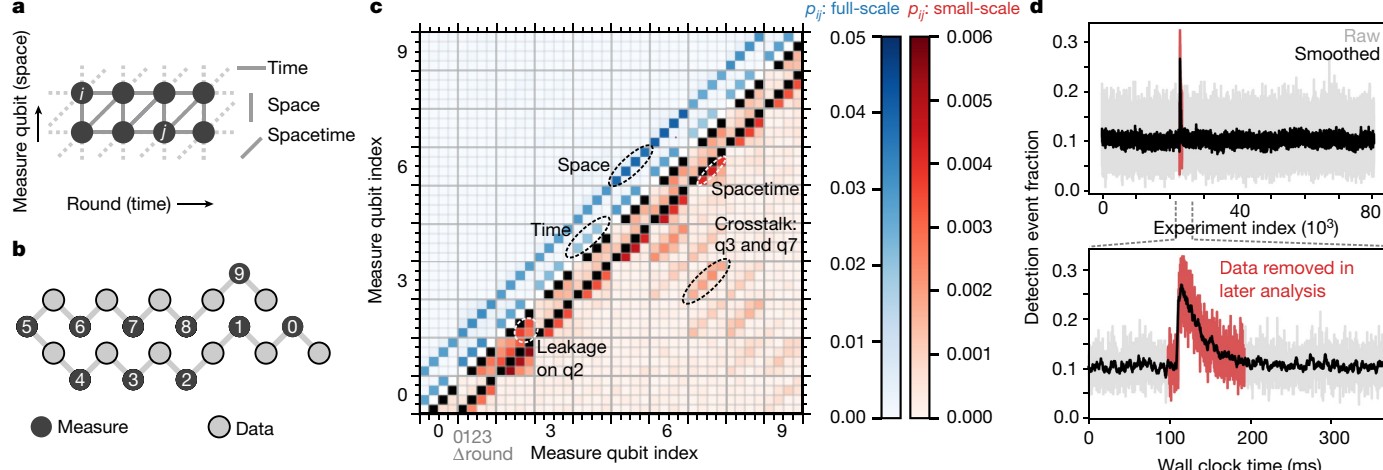

**Fig. 2 | Analysis of error detections. a**, Detection event graph. Errors in the code trigger two detections (except at the ends of the chain), each represented by a node. Edges represent the expected correlations due to data qubit errors (spacelike and spacetimelike) and measure qubit errors (timelike). **b**, Ordering of the measure qubits in the repetition code. **c**, Measured two-point correlations ($p_{ij}$) between detection events represented as a symmetric matrix. The axes correspond to possible locations of detection events, with major ticks marking measure qubits (space) and minor ticks marking difference ($\Delta$round) in rounds (time). For the purpose of illustration, we have averaged together the matrices for 4-round segments of the 50-round experiment shown in Fig. 1e and

section IX). Note that $p_{ij}$ is symmetric between $i$ and $j$. A description of the uncertainties on the matrix values can be found in Supplementary Information section IX. The upper triangle shows the full scale, where only the expected spacelike and timelike correlations are apparent. The lower triangle shows a truncated colour scale, highlighting unexpected detection pairs due to crosstalk and leakage. Note that observed crosstalk errors occur between next-nearest neighbours in the 2D array. **d**, Top: observed high-energy event in a time series of repetition code runs. Bottom: zoom-in on high-energy event, showing rapid rise and exponential decay of device-wide errors, and data that are removed when computing logical error probabilities.

section IX). Note that $p_{ij}$ is symmetric between $i$ and $j$. In Fig. 2c, we plot the $p_{ij}$ matrix for the data shown in Fig. 1e. In the upper triangle, we show the full scale of the data, where, as expected, the most visible correlations are either spacelike or timelike.

However, the sensitivity of this technique allows us to find features that do not fit the expected categories. In the lower triangle, we plot the same data but with the scale truncated by nearly an order of magnitude. The next most prominent correlations are spacetimelike, as we expect, but we also find two additional categories of correlations. First, we observe correlations between non-adjacent measure qubits in the same measurement round. Although these non-adjacent qubits are far apart in the repetition code chain, the qubits are in fact spatially close, owing to the embedding of the 1D chain in a 2D array. Optimization of gate operation frequencies mitigates crosstalk errors to a large extent[37], but suppressing these errors further is the subject of active research. Second, we find excess correlations between measurement rounds that differ by more than 1, which we attribute to leakage generated by a number of sources including gates[12] and thermalization[38,39]. For the observed crosstalk and leakage errors, the excess correlations are around $3 \times 10^{-3}$, an order of magnitude below the measured spacelike and timelike errors but well above the measurement noise floor of $2 \times 10^{-4}$.

Additionally, we observe sporadic events that greatly decrease performance for some runs of the repetition code. In Fig. 2d, we plot a time series of detection event fractions averaged over all measure qubits for each run of an experiment. We observe a sharp three-fold increase in detection event fraction, followed by an exponential decay with a time constant of 50 ms. These types of events affect less than 0.5% of all data taken (Supplementary Information section V), and we attribute them to high-energy particles such as cosmic rays striking the quantum processor and decreasing $T_1$ on all qubits[40,41]. For the purpose of understanding the typical behaviour of our system, we remove data near these events (Fig. 2d). However, we note that mitigation of these events through improved device design[42] and/or shielding[43] will be critical to implementing large-scale fault-tolerant computers with superconducting qubits.

## Logical errors in the repetition code

We decode detection events and determine logical error probabilities following the procedure in ref. [21]. Briefly, we use a minimum-weight perfect matching algorithm to determine which errors were most likely to have occurred given the observed detection events. Using the matched errors, we then correct the final measured state of the data qubits in post-processing. A logical error occurs if the corrected final state is not equal to the initial state. We repeat the experiment and analysis while varying the number of detection rounds from 1 to 50 with a fixed number of qubits, 21. We determine logical performance of smaller code sizes by analysing spatial subsets of the 21-qubit data (see Supplementary Section VII). These results are shown in Fig. 3a, where we observe a clear decrease in the logical error probability with increasing code size. The same data are plotted on a semilog scale in Fig. 3b, highlighting the exponential nature of the error reduction.

To extract logical error per round ($\varepsilon_L$), we fitted the data for each number of qubits (averaged over spatial subsets) to $2P_{error} = 1 - (1 - 2\varepsilon_L)^{n_{rounds}}$, which expresses an exponential decay in logical fidelity with number of rounds. In Fig. 3c, we show $\varepsilon_L$ for the phase-flip and bit-flip codes versus number of qubits used. We find more than 100× suppression in $\varepsilon_L$ for the phase-flip code from 5 qubits ($\varepsilon_L = 8.7 \times 10^{-3}$) to 21 qubits ($\varepsilon_L = 6.7 \times 10^{-5}$). Additionally, we fitted $\varepsilon_L$ versus code distance to equation (1) to extract $\Lambda$, and find $\Lambda_X = 3.18 \pm 0.08$ for the phase-flip code and $\Lambda_Z = 2.99 \pm 0.09$ for the bit-flip code.

## Error budgeting and projecting QEC performance

To better understand our repetition code results and project surface code performance for our device, we simulated our experiments with a depolarizing noise model, meaning that we probabilistically inject a random Pauli error ($X$, $Y$ or $Z$) after each operation (Supplementary Information section VIII). The Pauli error probabilities for each type of operation are computed using mean error rates and are shown in Fig. 4a. We first simulate the bit-flip and phase-flip codes using the error rates in Fig. 4a, obtaining values of $\Lambda$ that should be directly comparable to

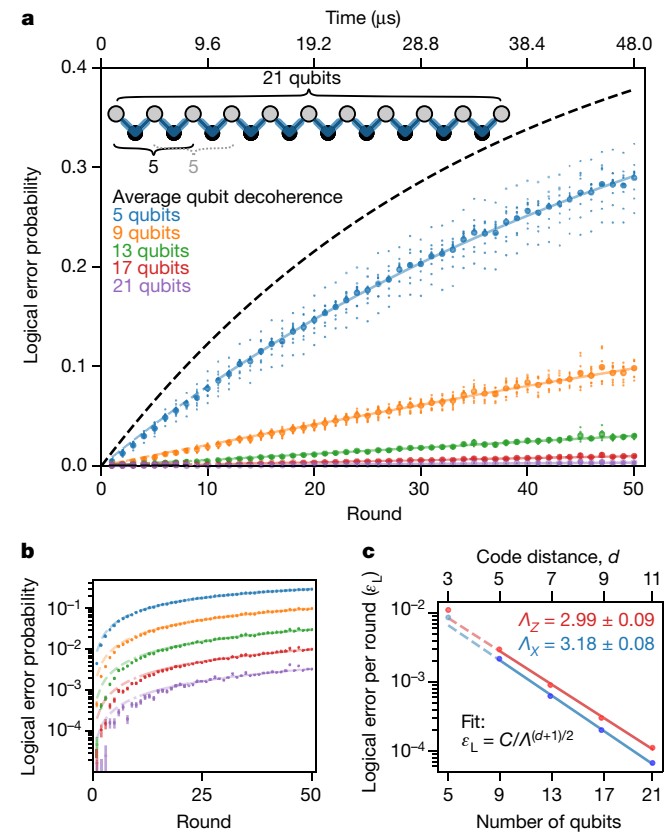

**Fig. 3 | Logical errors in the repetition code. a**, Logical error probability versus number of detection rounds and number of qubits for the phase-flip code. Smaller code sizes are subsampled from the 21-qubit code as shown in the inset; small dots are data from subsamples and large dots are averages. **b**, Semilog plot of the averages from **a** showing even spacing in log(error probability) between the code sizes. Error bars are estimated standard error from binomial sampling given the total number of statistics over all subsamples. The lines are exponential fits to data for rounds greater than 10. **c**, Logical error per round ($\varepsilon_L$) versus number of qubits, showing exponential suppression of error rate for both bit-flip and phase-flip, with extracted $\Lambda$ factors. The fits for $\Lambda$ and uncertainties were obtained using a linear regression on the log of the logical error per round versus the code distance. The fit excludes $n_{qubits} = 5$ to reduce the influence of spatial boundary effects (Supplementary Information section VII).

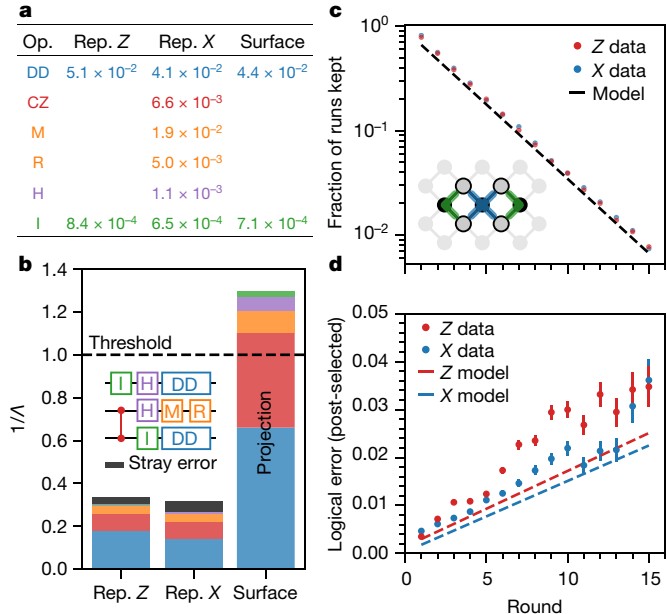

**Fig. 4 | Error budgeting repetition and surface codes. a**, Depolarizing error probability (bit flip errors for M and R) for various operations in the stabilizer circuit, derived from averaging quantities in Fig. 1b. Note that the idle gate (I) and dynamical decoupling (DD) values depend on the code being run because the data qubits occupy different states. Op., operation; Rep., repetition. **b**, Estimated error budgets for the bit-flip and phase-flip codes, and projected error budget for the surface code, based on the depolarizing errors from **a**. The repetition code budgets slightly underestimate the experimental errors, and the discrepancy is labelled stray error. For the surface code, the estimated $1/\Lambda$ corresponds to the difference in $\varepsilon_L$ between a $d = 3$ and $d = 5$ surface code, and is ~4 times higher than in the repetition codes owing to the more stringent threshold for the surface code. Rep., repetition **c**, For the $d = 2$ surface code, the fraction of runs that had no detection events versus number of rounds, plotted with the prediction from a similar error model as the repetition code (dashed line). Inset: physical qubit layout of the $d = 2$ surface code, seven qubits embedded in a 2D array. **d**, Probability of logical error in the surface code among runs with no detection events versus number of rounds. Depolarizing model simulations that do not include leakage or crosstalk (dashed lines) show good agreement. Error bars for **c** (not visible) and **d** are estimated standard error from binomial sampling with 240,000 experimental shots, minus the shots removed by post-selection in **d**.

our experimentally measured values. Then we repeat the simulations while individually sweeping the Pauli error probability for each operation type and observing how $1/\Lambda$ changes. The relationship between $1/\Lambda$ and each of the error probabilities is approximately linear, and we use the simulated sensitivity coefficients to estimate how much each operation in the circuit increases $1/\Lambda$ (decreases $\Lambda$).

The resulting error budgets for the phase-flip and bit-flip codes are shown in Fig. 4b. Overall, measured values of $\Lambda$ are approximately 20% worse than simulated values, which we attribute to mechanisms such as the leakage and crosstalk errors that are shown in Fig. 2c but were not included in the simulations. Of the modelled contributions to $1/\Lambda$, the dominant sources of error are the CZ gate and data qubit decoherence during measurement and reset. In the same plot, we show the projected error budget for the surface code, which has a more stringent threshold than the repetition code because the higher-weight stabilizers in both $X$ and $Z$ bases lead to more possible logical errors for the same code distance. We find that the overall performance of Sycamore must be improved to observe error suppression in the surface code.

Finally, we test our model against a distance-2 surface code logical qubit[19]. We use seven qubits to implement one weight-4 $X$ stabilizer

and two weight-2 $Z$ stabilizers as depicted in Fig. 1a. This encoding can detect any single error but contains ambiguity in mapping detections to corrections, so we discard any runs where we observe a detection event. We show the fraction of runs where no errors were detected in Fig. 4c for both logical $X$ and $Z$ preparations; we discard 27% of runs each round, in good agreement with the simulated prediction. Logical errors can still occur after post-selection if two or more physical errors flip the logical state without generating a detection event. In Fig. 4d, we plot the post-selected logical error probability in the final measured state of the data qubits, along with corresponding depolarizing model simulations. Linear fits of the experimental data give $2 \times 10^{-3}$ error probability per round averaged between the $X$ and $Z$ basis, while the simulations predict $1.5 \times 10^{-3}$ error probability per round. Supplementary Information section VI discusses potential explanations for the excess error in experiment, but the general agreement provides confidence in the projected error budget for surface codes in Fig. 4b.

## Conclusion and outlook

In this work, we demonstrate stable error detection event fractions while executing 50 rounds of stabilizer measurements on a Sycamore

device. By computing the probabilities of detection event pairs, we find that the physical errors detected on the device are localized in space and time to the $3 \times 10^{-3}$ level. Repetition code logical errors are exponentially suppressed when increasing the number of qubits from 5 to 21, with a total error suppression of more than $100\times$. Finally, we corroborate experimental results on both 1D and 2D codes with depolarizing model simulations and show that the Sycamore architecture is within a striking distance of the surface code threshold.

Nevertheless, many challenges remain on the path towards scalable quantum error correction. Our error budgets point to the salient research directions required to reach the surface code threshold: reducing CZ gate error and data qubit error during measurement and reset. Reaching this threshold will be an important milestone in quantum computing. However, practical quantum computation will require $\Lambda \approx 10$ for a reasonable physical-to-logical qubit ratio of 1,000:1 (Supplementary Information section VI). Achieving $\Lambda \approx 10$ will require substantial reductions in operational error rates and further research into mitigation of error mechanisms such as high-energy particles.

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

**Google Quantum AI**

Zijun Chen[1], Kevin J. Satzinger[1], Juan Atalaya[1], Alexander N. Korotkov[1,2], Andrew Dunsworth[1], Daniel Sank[1], Chris Quintana[1], Matt McEwen[1,3], Rami Barends[1], Paul V. Klimov[1], Sabrina Hong[1], Cody Jones[1], Andre Petukhov[1], Dvir Kafri[1], Sean Demura[1], Brian Burkett[1], Craig Gidney[1], Austin G. Fowler[1], Alexandru Paler[4,5], Harald Putterman[1,10], Igor Aleiner[1], Frank Arute[1], Kunal Arya[1], Ryan Babbush[1], Joseph C. Bardin[1,6], Andreas Bengtsson[1], Alexandre Bourassa[1,7], Michael Broughton[1], Bob B. Buckley[1], David A. Buell[1], Nicholas Bushnell[1], Benjamin Chiaro[1], Roberto Collins[1], William Courtney[1], Alan R. Derk[1], Daniel Eppens[1], Catherine Erickson[1], Edward Farhi[1], Brooks Foxen[1], Marissa Giustina[1], Ami Greene[1,8], Jonathan A. Gross[1], Matthew P. Harrigan[1], Sean D. Harrington[1], Jeremy Hilton[1], Alan Ho[1], Trent Huang[1], William J. Huggins[1], L. B. Ioffe[1], Sergei V. Isakov[1], Evan Jeffrey[1], Zhang Jiang[1], Kostyantyn Kechedzhi[1], Seon Kim[1], Alexei Kitaev[1,9], Fedor Kostritsa[1], David Landhuis[1], Pavel Laptev[1], Erik Lucero[1], Orion Martin[1], Jarrod R. McClean[1], Trevor McCourt[1], Xiao Mi[1], Kevin C. Miao[1], Masoud Mohseni[1], Shirin Montazeri[1], Wojciech Mruczkiewicz[1], Josh Mutus[1], Ofer Naaman[1], Matthew Neeley[1], Charles Neill[1], Michael Newman[1], Murphy Yuezhen Niu[1], Thomas E. O'Brien[1], Alex Opremcak[1], Eric Ostby[1], Bálint Pató[1], Nicholas Redd[1], Pedram Roushan[1], Nicholas C. Rubin[1], Vladimir Shvarts[1], Doug Strain[1], Marco Szalay[1], Matthew D. Trevithick[1], Benjamin Villalonga[1], Theodore White[1], Z. Jamie Yao[1], Ping Yeh[1], Juhwan Yoo[1], Adam Zalcman[1], Hartmut Neven[1], Sergio Boixo[1], Vadim Smelyanskiy[1], Yu Chen[1], Anthony Megrant[1✉] & Julian Kelly[1✉]

[1]Google LLC, Mountain View, CA, USA. [2]Department of Electrical and Computer Engineering, University of California, Riverside, CA, USA. [3]Department of Physics, University of California, Santa Barbara, CA, USA. [4]Johannes Kepler University, Linz, Austria. [5]University of Texas at Dallas, Richardson, TX, USA. [6]Department of Electrical and Computer Engineering, University of Massachusetts, Amherst, MA, USA. [7]Pritzker School of Molecular Engineering, University of Chicago, Chicago, IL, USA. [8]Research Laboratory of Electronics, Massachusetts Institute of Technology, Cambridge, MA, USA. [9]California Institute of Technology, Pasadena, CA, USA. [10]Present address: AWS Center for Quantum Computing, Pasadena, CA, USA. ✉e-mail: amegrant@google.com; juliankelly@google.com

## Methods

### The Sycamore processor

In this work, we use a Sycamore quantum processor consisting of 54 superconducting transmon qubits and 88 tunable couplers in a 2D array. The available operational frequencies of the qubits range from 5 GHz to 7 GHz. The couplers are capable of tuning the qubit–qubit couplings between 0 MHz and 40 MHz, allowing for fast entangling gates while also mitigating unwanted stray interactions. The qubits and couplers in the Sycamore processor are fabricated using aluminium metallization and aluminium/aluminium-oxide Josephson junctions. Indium bump bonds are used to connect a chip containing control circuitry to the chip containing the qubits. The hybridized device is then wire-bonded to a superconducting circuit board and cooled below 20 mK in a dilution refrigerator.

### Control and readout

Each qubit is connected to a microwave control line used to drive *XY* rotations, while qubits and couplers are each connected to flux control lines that tune their frequencies and are used to perform CZ and reset operations. Additionally, each qubit is coupled to a resonator with frequency around 4.5 GHz for dispersive readout, and six such resonators are frequency multiplexed and coupled to a microwave transmission line via a common Purcell filter. Microwave drive and flux lines are connected via multiple stages of wiring and filters to arbitrary waveform generators (AWGs) at room temperature. The AWGs for both microwave and flux control operate at 1 gigasample per second, and for the microwaves, signals are additionally upconverted with single sideband mixing to reach the qubit frequencies. The outputs of the readout transmission lines are additionally connected to a series of amplifiers—impedance matched parametric amplifiers at 20 mK, high-electron-mobility transistor amplifiers at 3 K, and room-temperature amplifiers—before terminating in a downconverter and analogue–digital converter (ADC). Low-level operation of the AWGs is controlled by FPGAs. Construction and upload of control waveforms and discrimination of ADC signals are controlled by classical computers running servers that each control different types of equipment, and a client computer that controls the overall experiment.

### Calibration

Upon initial cooldown, various properties of each qubit and coupler (including coherence times as a function of frequency, control couplings, and couplings between qubits and couplers) are characterized individually. An optimizer is then used to select operational frequencies for gates and readout for each qubit (or pair of qubits for the CZ gate). The optimizer's objective function is the predicted fidelity of gate operations and is designed to incorporate coherence times, parasitic couplings between qubits, and microwave non-idealities such as crosstalk and carrier bleedthrough. More information about the optimization can be found in refs. [33,37] and in Supplementary Information section XII. Next, the primary operations required for QEC (SQ gates, CZ, reset, readout) are calibrated individually. Finally, we perform a round of QEC specific calibrations for phase corrections (see Supplementary Information section III). Automated characterizations and calibrations are described using a directed acyclic graph, which determines the flow of experiments from basic characterizations to fine tuning[44].

### Execution of the experiment

Circuits for the repetition codes and *d* = 2 surface code were specified using Cirq[45], then translated into control waveforms based on calibration data. The exact circuits that were run are available on request. For the bit-flip and phase-flip repetition codes, the 80,000 total experimental shots for each number of rounds were run in four separate experiments.

Each experiment consisted of randomly selecting initial data qubit states, running for 4,000 shots, then repeating that process five times for 20,000 shots total. In between shots of the experiment, the qubits idle for 100 μs and are also reset. The 400 total experiments (one bit-flip and one phase-flip code for each total number of error correction rounds between 1 and 50, and four experiments for each number of rounds) were shuffled before being run. Data for the distance-2 surface code was similarly acquired, but with 15,000 shots for each of the 16 possible data qubit states for 240,000 shots total, and shuffling was done within each number of rounds over the data qubit states, but no shuffling was done over the number of rounds or data qubit basis.

### Data analysis

As described in the main text, for each experimental shot, the array of raw parity measurements is first prepended with initial data qubit parities and appended with final measured data qubit parities. Then the parity values are turned into an array detection events by computing the XOR between each neighbouring round of measurements, resulting in an array that is one less in the 'rounds' dimension. For the repetition code data, cosmic rays are post-selected by first computing the total detection event fraction for each experimental shot, producing an array of 80,000 values between 0 and 1. Next, we apply a moving average to that array, with a rectangular window of length 20. Finally, we find where the moving average exceeds 0.2 and remove 100 shots before crossing the threshold and 600 shots following the crossing of the threshold. The analysis then proceeds through minimum-weight perfect matching and exponential fits of logical error rate per round and Λ, as described in the main text and in more detail in Supplementary Section X. Cosmic ray post-selection is not done for the *d* = 2 surface code data, since the analysis as described in the main text already post-selects any shots where errors are detected.

## Data availability

The data that support the plots within this paper and other findings of this study are available from the corresponding authors upon reasonable request.

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

**Acknowledgements** We thank J. Platt, J. Dean and J. Yagnik for their executive sponsorship of the Google Quantum AI team, and for their continued engagement and support. We thank S. Leichenauer and J. Platt for reviewing a draft of the manuscript and providing feedback.

**Author contributions** Z.C., K.J.S., H.P., A.G.F., A.N.K. and J.K. designed the experiment. Z.C., K.J.S. and J.K. performed the experiment and analysed the data. C.Q., K.J.S., A. Petukhov and Y.C. developed the controlled-*Z* gate. M. McEwen, D.K., A. Petukhov and R. Barends developed the reset operation. M. McEwen and R. Barends performed experiments on leakage, reset and high-energy events in error correcting codes. D. Sank and Z.C. developed the readout operation. A.D., B.B., S.D. and A.M. led the design and fabrication of the processor. J.A. and A.N.K. developed and performed the $p_{ij}$ analysis. C.J. developed the inverse Λ model and performed the simulations. A.G.F. and C.G. wrote the decoder and interface software. S. H., K.J.S. and J.K. developed the dynamical decoupling protocols. P.V.K. developed error mitigation techniques based on system frequency optimization. Z.C., K.J.S., S.H., P.V.K. and J.K. developed error correction calibration techniques. Z.C., K.J.S. and J.K. wrote the manuscript. S.B., V. Smelyanskiy, Y.C., A.M. and J.K. coordinated the team-wide error correction effort. Work by H. Putterman was done prior to joining AWS. All authors contributed to revising the manuscript and writing the Supplementary Information. All authors contributed to the experimental and theoretical infrastructure to enable the experiment.

**Competing interests** The authors declare no competing interests.

**Additional information**
**Correspondence and requests for materials** should be addressed to A.M. or J.K.
