## [Peer Review File · Nature]

Manuscript Title: Exponential suppression of bit or phase errors with cyclic error correction

Reviewer Comments & Author Rebuttals

Reviewer Reports on the Initial Version:

Referees' comments:

Referee #1 (Remarks to the Author):

This is a breakthrough paper in the field of quantum error correction and towards demonstrating fault tolerant quantum computing. The paper reports for the first time an exponential suppression of bit- and phase-flip errors using cyclic stabilizer measurements. This experiment is also relevant for analyzing error correlations, locality of errors and explore error mitigation techniques. The presentation is clear, the main text is well supported by the more detailed explanations in the supplementary information document, and quality of data is very good. I thus, recommend its publication in Nature.

Some points to be addressed:

- As mentioned in the manuscript, at most 21 of the 54 qubits of the Sycamore processor were used for implementing both the distance-11 repetition code and the distance-2 surface code. Assuming not all qubits (and couplings) behave in the same way, I guess the most reliable qubits of the processor were selected for running the experiments. I think it would be valuable if the authors could elaborate on the choice of the qubits. How the variability among qubits would affect the results of the experiment?
- Different types of unexpected error correlations are observed due to crosstalk, leakage and high energy events. It can be observed in Figure 2c that correlations caused by leakage and crosstalk are of similar order of magnitude (10^{-3}), but how do they compare with the correlated errors introduced by high energy events? It would be possible to show them also in Fig. 2c or in a similar graph? And could the authors briefly introduced potential techniques to mitigate this kind of errors?
- In Fig. 4d the logical error rate for the X basis is slightly lower than for the Z basis. Why is that? And in the same graph, why for the simulated date is the other way around (i.e. X model logical error rate is higher than the Z model)?
- For the simulation, the depolarizing error model was used. According to the authors, the main reason for that is that it allows to simulate larger code distances. However, this model underestimates the logical error rate as crosstalk and leakage are not considered and as discussed in Section III they are important sources of errors. It would be good to show simulations using a more accurate error model including both sources of errors at least for distance-2 surface code. What is the limit in number of quits that can be simulated under a more realistic noise model?
- In Figure S3a, the fraction of data that was discarded for each run of the repetition code is shown. It is difficult to analyze the data in the way it is represented. I suggest using a different representation (e.g. bar diagram). I would be good if the authors could elaborate on the observed distribution. In addition, it is also difficult to compare the logical error rate probabilities if the energy events are kept with the ones in which they are discarded. A solution would be to plot both

in the same graph or discuss their differences in the text.

- Figure S5 shows the quantum circuit for distance-2 surface code. I assume that gates vertically aligned are performed in parallel. Is there any reason for performing some Hadamards at a specific time-step (i.e. vertically aligned gates). For instance, the first H on q1 is applied in the first step and could be applied at any step before the CZ. The same applies to Hs on the top and bottom qubits. How does the order of gates affect the performance of the circuit? If it is affected by the gate scheduling, why is that (i.e. crosstalk, leakage...)?

- To decode the errors obtained in the experiment, the authors use the very popular minimum-weight perfect matching algorithm. They tried different strategies to determine the expected edge probabilities and weights. This is important because this will affect the accuracy of the decoder and therefore will impact the resulting logical error rate. Have the authors considered to use a different decoding algorithm? Or even a neural network-based decoder?

Some typos to be corrected:

- In Section I of the supplementary information there is a ref to Fig S3 (end of first page) that I think should be to Fig. S2.

- In Fig. S3 the text of the caption is missing a reference to a Figure (Fig. ?).

Referee #2 (Remarks to the Author):

A fault-tolerant universal quantum computer is the ultimate vision of the current endeavor to build sophisticated controllable quantum devices that can compute problems that are otherwise intractable, or simulate physical quantum systems that cannot be simulated by classical computers. One of the most important ingredients of such a universal quantum machine is error correction: Many physical qubits taken together can be made to act as one or more logical qubits that are less sensitive to errors than the underlying physical qubits themselves. For this to work, we need on the one hand high-quality qubits and high-fidelity gate operations, and on the other hand an error identification scheme that allows us to detect errors without destroying the quantum state itself. Although current technology has tremendously improved over the last years and decades, current quantum processors are still not capable of running algorithms on logical qubits, mainly because decoherence sets in still too early. Moreover, crosstalk and correlated errors make it difficult to control the qubits at the required level across an entire qubit array.

In the manuscript 'Exponential suppression of bit or phase flip errors with repetitive error correction' the Google team demonstrates that by extending the code distance to more and more physical qubits that errors can be exponentially suppressed in a repetition code. They use a Sycamore processor with improved performance over earlier version that comes from improved design of the readout. Unfortunately, they only state that the device has improved but do not give any details about the particular nature of the improvement. Providing details also on the technical level would definitely further increase the relevance of the manuscript for the community working on QC hardware, instead of demonstrating— admittedly, though, in an impressive manner – a specific error detection algorithm alone. It would also be interesting to learn more about the feasibility for error correction. What are the challenges to correct the errors immediately, and what

will be the effect on the overall error when including a correction cycle?

The authors show that they can carry out up to 50 consecutive error detection cycles to identify both bit flip and phase flip errors, albeit not at the same time. Using an increasing number of qubits they find both for the bit and the phase flip code an exponential suppression. They also demonstrate a small 2D surface code implementation and argue that this demonstrates the forward compatibility of their architecture. Unfortunately, they couldn't demonstrate yet the exponential suppression for the surface code. In this respect, I am wondering how the different error thresholds of the repetition code and the surface code affect the result. In the repetition code the physical qubit error threshold is as high as 0.5, while it is much lower in the surface code. The authors should explain in more details the role of the error threshold when they write about the exponential scaling 'below a certain threshold determined by the decoder' in the introduction. Moreover, Shor's code could as a simple extension to the repetition code correct for arbitrary qubit error and should in principle be readily realizable on the Sycamore processor. What would be the estimated performance of this code?

The authors carefully study both temporal and spatial correlations of errors, i.e. errors that occur on data qubits and measurement qubits. As an important technical result, they find that most of the errors are local. With their method, the authors identify also smaller correlated errors caused either by crosstalk between qubits that are physically close on a chip, or by long-lived correlations that they attribute to leakage or gate errors.

Interestingly they also identify device-wide correlated errors with decay time of several tens of milliseconds. Does this mean that the device is unusable for this duration of time? It would be interesting to learn about the temporal distribution of these events, i.e. if these follow, e.g., a Poissonian distribution or not. Is it conceivable that in a realistic mid- to large scale QC system these events (experiments) can be eliminated via post-selection, which would then only affect the effective clock-rate of the processor. Or is the only way forward to find mitigation strategies that overcome these device-wide errors?

Furthermore, the Google team carefully analyzes the contribution of each operation on the overall error. They conclude that the dominant error sources are the two-qubit CZ gate and decoherence of the qubits during idle time, despite the use of dynamical decoupling sequences. While the authors diligently discuss the effect of errors, I would encourage the authors to include a more detailed discussion of the physical nature of these errors in the text.

In conclusion, while not shown for the surface code the authors demonstrate impressively that by extending the code size of the repetition code that errors are exponentially suppressed, a result that may definitely warrant publication in a highly visible journal such as Nature. By providing more details on specific points (error threshold, details on improvements of the chip, discussion of noise sources), the manuscript could, however, serve not only as a mere demonstration of what can be technically done, but also as a perspective for the community what measures have to be taken towards further improved capabilities.

Referee #3 (Remarks to the Author):

In the article 'exponential suppression of bit or phase flip errors with repetitive error correction' the authors report on an experiment using the Sycamore chip where they demonstrate logical error suppression with repeated stabilizer measurements of repetition codes of varying distances.

I have been asked to respond to the following:

1. "Does the manuscript have flaws which should prohibit its publication?"
2. "If the conclusions are not original, it would be very helpful if you could provide relevant references."
3. "Do you feel that the results presented are of immediate interest to many people in your own discipline, or to people from several disciplines?"

and

4. "If you recommend publication, please outline what you consider to be the outstanding features"

I think this is excellent work. While I have some issues with the manuscript that I will detail below I believe the results suitable for publication in Nature. As such I will answer 1 last, with my list of criticisms.

In answer to 2, I think these conclusions are original. The authors give a detailed list of related work on the final page of the supplemental material which I believe is fairly extensive, although I have some more additions to the list in my comments below. The work here goes beyond the state of the art in this list.

In answer to 3 and 4 together - This work is exceptional in the sense that it has combined many of the necessary elements needed for a basic demonstration of error correction. They have achieved sufficient control over their system to show that they can suppress the logical failure rate by growing the system size of a repetition code. This step is essential to the realisation of scalable quantum computation. The work will be of interest to the experimental community that are developing controlled quantum systems for quantum computation, and will impact the theoretical community working to design scalable quantum computing architectures. While the system is not sufficiently well controlled yet to realise a scalable surface code, the researchers have made significant strides towards this important goal. I believe this warrants publication.

Finally, let me answer 1 with some changes I would like to see addressed before recommending publication.

* I don't agree with the conclusion on page 3, where it is suggested that cross talk errors are short range. The conclusion is made by identifying a short-range cross-talk error "...which suggests that while crosstalk exists in our system, it is short range.". However, measuring some short-range crosstalk is not evidence that there is no long range crosstalk noise in the system. To make this conclusion more experiments would need to be conducted using more qubits of the chip.

* I would also suggest improving the final sentence of the abstract. I don't think it should be too hard to write a stronger sentence that reflects the results of the paper better. I say this because I think superconducting qubits have always been a 'viable path' towards fault tolerant quantum computing. In this sense I would say that this sentence as it is is a little vacuous. As you say in your conclusions, you are still approaching the threshold error rate to achieve scalable 2D codes. This is the state of the art in most experimental set ups at the moment, so anyone who has demonstrated a small system of interacting qubits could make a similar claim. Nevertheless, there has been a lot of progress made in this work. I think a more specific statement will highlight the importance of the work.

* One of my main problems is that in general I find the writing to be very lazy. Many sentences are ambiguous, too colloquial, poorly worded, and/or could be interpreted as incorrect. For example, the second sentence of the introduction seems to imply that measure qubits are unphysical, whereas clearly they are physical qubits too. The first sentence is also very strange.

Stabilizer codes are just subspaces of a larger Hilbert space. I don't think an 'error-correction scheme' is a well-defined technical term, but to me it implies, among other things, hardware with stabilizer readout circuits and a decoder that carry out the dynamics of quantum error correction. Therefore, the opening statement 'Many quantum error correction schemes can be classified as stabilizer codes,...' does not make very much sense to me. While the introduction seems to be particularly bad, sentences like these continue throughout the manuscript. Given the number of coauthors, I believe that this work could have been written up with much more care. I am unwilling to recommend publication unless some effort is made to improve the writing substantially.

Some other comments.

* In the abstract, I don't agree with the statement 'QEC also requires that the errors are local...'. One could come up with a set of non-local errors that are correctable with some code according to the Knill-Laflamme error correction conditions. I understand that local errors are important for the protocol presented here, but I think this statement could be made more precise.

* I think it is important that the original Dennis et al. paper is cited as an original work introducing the surface code, and for introducing the minimum-weight perfect matching decoder, the method of decoding used here. Likewise Kitaev's original 'Fault tolerant quantum computing with anyons' paper should be cited as the seminal work where the toric code is introduced.

* In the caption for figure 4d it says the data shows good agreement with the model, although the data seems to diverge from the straight lines that are plotted. In the main text it is claimed that this is due to crosstalk and leakage, but perhaps this should be mentioned in the caption and/or marked on the figure as well. I see that more details are given in the supplemental material to explain this discrepancy, but is it possible to calculate how many errors are due to leakage and crosstalk to a leading order by comparing the model with a straight line fitted to the data?

* You should add references [Harper and Flammia, PRL 122, 080504 (2019)] and [Willsch et al. Phys. Rev. A 98, 052348 (2018)] to your list of references in the final table in your supplemental material, and perhaps the work of [Heeres et al. Nat. Commun. 8, 94 (2017)] could be included too.

Author Rebuttals to Initial Comments:

Referee 1

We would like to express our sincere gratitude to the referee for taking their time to review our manuscript and for recommending our work for publication in Nature. Below we describe improvements made to the manuscript as suggested by the referee.

"As mentioned in the manuscript, at most 21 of the 54 qubits of the Sycamore processor were used for implementing both the distance-11 repetition code and the distance-2 surface code. Assuming not all qubits (and couplings) behave in the same way, I guess the most reliable qubits of the processor were selected for running the experiments. I think it would be valuable if the authors could elaborate on the choice of the qubits. How the variability among qubits would affect the results of the experiment?"

21 qubits were chosen out of 30 qubits which were actually wired up in the cryostat. We have added the following text to the main text:

“Like its predecessor, this processor has 54 qubits, but we used at most 21 because only a subset of the processor was wired-up.”

The referee correctly points out that variability of the qubits played a role in selection of the exact repetition code chain. We have added a section to the supplement (now Section I) which discusses the primary improvement in this Sycamore device, which was the reduction in resonator ringdown times, and it includes the following sentence:

“While this new Sycamore generation has tighter spread in resonator ringdown times, operation times for QEC are still limited by the slowest measure qubit readout resonator. The specific chain of 21 qubits used in the repetition code was chosen to minimize the longest resonator ringdown time among measure qubits.”

“Different types of unexpected error correlations are observed due to crosstalk, leakage and high energy events. It can be observed in Figure 2c that correlations caused by leakage and crosstalk are of similar order of magnitude (10^{-3}), but how do they compare with the correlated errors introduced by high energy events? It would be possible to show them also in Fig. 2c or in a similar graph? And could the authors briefly introduce potential techniques to mitigate this kind of errors?”

In the original version, the word “correlation” was used in two different senses: first, to refer to the pairwise correlated probabilities in the p_{ij} matrix, and second to say that the entire device experience heightened errors during a high energy event. We have significantly rewritten the paragraph on high energy events to avoid this confusion. Nevertheless, we addressed the referee’s specific request in Supplement Section V, where we now compare p_{ij} analysis between a high energy event and baseline, showing increased background levels of pairwise correlation between all qubits on the device. Additionally, we explicitly mention mitigation techniques in the main text along with citations:

“However, we note that mitigation of these events via improved device design \cite{karatsu2019mitigation} and/or shielding \cite{cardani2020reducing} will be critical to implementing large-scale fault-tolerant computers with superconducting qubits. “

“In Fig. 4d the logical error rate for the X basis is slightly lower than for the Z basis. Why is that? And in the same graph, why for the simulated data is the other way around (i.e. X model logical error rate is higher than the Z model)?”

First, we would like to thank the referee for their attention to detail. In revisiting this figure, we realized that we had made an error in the colors of the plot, and in fact, the ordering of X vs Z agrees between experiment and simulation. We apologize for this confusion, and the

figure has now been fixed.

Second, in supplement Section VI we have added a new paragraph starting with “**Note that in the simulation, the XX basis has slightly lower error than the ZZ basis**”. Briefly, there is an asymmetry in the $d=2$ code between having 2 weight-2 Z stabilizers vs 1 weight-4 X stabilizer, which may cause lower error in the X basis. We also provide a few potential explanations for the numerical differences between simulation and experiment, such as a bias in X vs Z noise and increased sensitivity to phase errors.

For the simulation, the depolarizing error model was used. According to the authors, the main reason for that is that it allows to simulate larger code distances. However, this model underestimates the logical error rate as crosstalk and leakage are not considered and as discussed in Section III they are important sources of errors. It would be good to show simulations using a more accurate error model including both sources of errors at least for distance-2 surface code. What is the limit in number of qubits that can be simulated under a more realistic noise model?

The limiting factor as it pertains to this work is not the scale of the computation required for the simulation, but rather, a deep enough understanding of the physical origins and dynamics of these “non-standard” error processes to be modeled accurately enough. Since the submission of this work we have made strides in, for example, understanding how leakage propagates between qubits in the Sycamore architecture, but we do not feel that these results are ready for publication yet. We have added the following sentence to Supplement Section VIII:

Additionally, even crude approximations of effects such as leakage require physical understanding of the origins and dynamics of these errors, and both experimental and theoretical work are ongoing in this direction.

In Figure S3a, the fraction of data that was discarded for each run of the repetition code is shown. It is difficult to analyze the data in the way it is represented. I suggest using a different representation (e.g. bar diagram). It would be good if the authors could elaborate on the observed distribution. In addition, it is also difficult to compare the logical error rate probabilities if the energy events are kept with the ones in which they are discarded. A solution would be to plot both in the same graph or discuss their differences in the text.

We thank the referee for the suggestions and have implemented both of them in Fig S4 and Fig. S5. With regards to the temporal distribution of the high energy events, more rigorous studies involving longer experiments, more careful timing, and more statistics are underway within our group and will be published in a future dedicated publication.

“Figure S5 shows the quantum circuit for distance-2 surface code. I assume that gates vertically aligned are performed in parallel. Is there any reason for performing some Hadamards at a specific time-step (i.e. vertically aligned gates). For instance, the first H on q1 is applied in the first step and could be applied at any step before the CZ. The same applies to Hs on the top and

bottom qubits. How does the order of gates affect the performance of the circuit? If it is affected by the gate scheduling, why is that (i.e. crosstalk, leakage...)?”

The referee correctly points out a valid micro-optimization that we could have done, but we did not get a chance to optimize the circuit for this publication. The circuit as implemented was convenient for software scheduling purposes. That said, we don't think the placement of Hadamards was important to the performance since microwave gates are not the dominant source of crosstalk or leakage.

To decode the errors obtained in the experiment, the authors use the very popular minimum-weight perfect matching algorithm. They tried different strategies to determine the expected edge probabilities and weights. This is important because this will affect the accuracy of the decoder and therefore will impact the resulting logical error rate. Have the authors considered to use a different decoding algorithm? Or even a neural network-based decoder?

Different decoding algorithms are indeed an active area of research within our team, but we have no results to report as it pertains to this work. We have added the following sentence to Supplement Section X:

Other potential techniques for decoding detections include maximum likelihood \cite{terhal2015quantum} and neural networks \cite{liu2019neural, varsamopoulos2017decoding}. The efficacy of these other methods compared to minimum weight perfect matching is currently under investigation.

Some typos to be corrected:

- In Section I of the supplementary information there is a ref to Fig S3 (end of first page) that I think should be to Fig. S2.*
- In Fig. S3 the text of the caption is missing a reference to a Figure (Fig. ?).*

These issues have been fixed.

Referee 2

We would like to express our sincere gratitude to the referee for taking their time to review our manuscript, and for their interest in having more technical details. Below we describe improvements made to the manuscript as suggested by the referee.

Providing details also on the technical level would definitely further increase the relevance of the manuscript for the community working on QC hardware, instead of demonstrating– admittedly, though, in an impressive manner – a specific error detection algorithm alone.

We agree with the referee and have added three new Supplement Sections: Section I, on

improvements to the Sycamore device and Section II, on error budgets for the CZ gate, and Section III, on a crucial calibration technique used in the experiment, to address this concern.

It would also be interesting to learn more about the feasibility for error correction. What are the challenges to correct the errors immediately, and what will be the effect on the overall error when including a correction cycle?

One of the advantages of the surface code is that correction of physical errors at the hardware level is not necessary for quantum memory. We have added the following sentence to the main text:

For the purpose of maintaining a logical quantum memory in the codes presented in this work, these errors can be compensated in classical software \cite{fowler2012surface}.

However, the referee raises a salient point that logical feedforward will be necessary to execute non-Clifford gates via state distillation, an important component of universal quantum computation, but it is not the focus of this work.

I am wondering how the different error thresholds of the repetition code and the surface code affect the result. In the repetition code the physical qubit error threshold is as high as 0.5, while it is much lower in the surface code. The authors should explain in more details the role of the error threshold when they write about the exponential scaling ‘below a certain threshold determined by the decoder’ in the introduction.

We thank the referee for raising this discussion as we think it is valuable to expand upon in our paper. We have made numerous changes to the main text.

In the introduction, first, we acknowledge that many factors play into the value of a single parameter threshold:

“In the simplest model, if the physical error per operation p is below a certain threshold p_{th} determined by **quantum computer architecture, chosen QEC code, and decoder**”

Second, we highlight that the “threshold” in real devices cannot be easily characterized by a simple number, rather we must measure the actual exponential suppression factor in the hardware:

“More realistic error models cannot be characterized by a single error rate p or a single threshold value p_{th} . Instead, quantum processors must be benchmarked by measuring Λ .”

In Section 4, we have added a short discussion on why the surface code threshold is more stringent than the repetition code threshold, as evidenced by the fact that λ is smaller ($1/\lambda$ is larger) in the surface code compared to the repetition code given the same

error model.

“In the same plot, we show the projected error budget for the surface code, which has a more stringent threshold than the repetition code because the higher-weight stabilizers in both X and Z bases lead to more possible logical errors for the same code distance. “

And finally, in the caption for Fig. 4b, we add a numerical estimate for “how much more stringent”

“...the estimated $1/\Lambda$ corresponds to the difference in ϵ_L between a $d=3$ and $d=5$ surface code, and is ~ 4 times higher than for the repetition codes due to the more stringent threshold for the surface code.”

Note that a repetition code has a threshold of 0.5 in a model where the only error process is bit-flip (or phase-flip) errors on the data qubits, which is not applicable to our system.

Moreover, Shor's code could as a simple extension to the repetition code correct for arbitrary qubit error and should in principle be readily realizable on the Sycamore processor. What would be the estimated performance of this code?

To our knowledge, Shor's code involves weight-6 stabilizers which would be difficult to measure repeatedly within our architecture.

Interestingly they also identify device-wide correlated errors with decay time of several tens of milliseconds. Does this mean that the device is unusable for this duration of time? It would be interesting to learn about the temporal distribution of these events, i.e. if these follow, e.g., a Poissonian distribution or not. Is it conceivable that in a realistic mid- to large scale QC system these events (experiments) can be eliminated via post-selection, which would then only affect the effective clock-rate of the processor. Or is the only way forward to find mitigation strategies that overcome these device-wide errors?

The device is in principle usable during these high energy events, however, as suggested in Vepsalainen et al (2020), the T_1 times of all the qubits are damped significantly. We now explicitly state this in the main text.

We agree that learning about the temporal distribution of these events would be interesting, but would require a more carefully timed experiment than our QEC experiments with significantly more statistics. This study is currently underway and will appear in a dedicated publication.

Regarding the idea of post-selection, the referee brings up an interesting idea. We think that the viability of the proposal would depend on the duration of the logical computation, and that for certain computations of interest which require potentially hours or days of runtime (e.g. Shor's algorithm), at least some mitigation will be required. We now explicitly call out potential mitigation strategies in the edited line:

“However, we note that mitigation of these events via improved device design \cite{karatsu2019mitigation} and/or shielding \cite{cardani2020reducing} will be critical to implementing large-scale fault-tolerant computers with superconducting qubits. “

Furthermore, the Google team carefully analyzes the contribution of each operation on the overall error. They conclude that the dominant error sources are the two-qubit CZ gate and decoherence of the qubits during idle time, despite the use of dynamical decoupling sequences. While the authors diligently discuss the effect of errors, I would encourage the authors to include a more detailed discussion of the physical nature of these errors in the text.

We have added a new supplemental section (Section II) discussing the error contributions when operating the CZ gate both in isolation and simultaneously. The white-noise dephasing (that is, the leftover dephasing after dynamical decoupling which is not explained by $2\pi T_1$) has not been carefully error budgeted.

Referee 3

We would like to express our sincere gratitude to the referee for taking their time to review our manuscript, and for stating that the results are suitable for publication in Nature. Below we describe improvements made to the manuscript as suggested by the referee.

I don't agree with the conclusion on page 3, where it is suggested that cross talk errors are short range. The conclusion is made by identifying a short-range cross-talk error "...which suggests that while crosstalk exists in our system, it is short range.". However, measuring some

short-range crosstalk is not evidence that there is no long range crosstalk noise in the system. To make this conclusion more experiments would need to be conducted using more qubits of the chip.

We agree, and have reworded the sentence as follows so that it makes no general claim about crosstalk in our device.

While these non-adjacent qubits are far apart in the repetition code chain, the qubits are in fact spatially close due to the embedding of the 1D chain in a 2D array.

I would also suggest improving the final sentence of the abstract. I don't think it should be too hard to write a stronger sentence that reflects the results of the paper better. I say this because I think superconducting qubits have always been a 'viable path' towards fault tolerant quantum computing. In this sense I would say that this sentence as it is a little vacuous. As you say in your conclusions, you are still approaching the threshold error rate to achieve scalable 2D codes. This is the state of the art in most experimental set ups at the moment, so anyone who has demonstrated a small system of interacting qubits could make a similar claim. Nevertheless, there has been a lot of progress made in this work. I think a more specific statement will highlight the importance of the work.

We thank the referee to encouraging us to highlight the significance of our work, and the final sentence of the abstract now reads:

“The successful demonstration of exponential error suppression over many rounds provides a foundation for building a scalable fault-tolerant quantum computer with superconducting qubits.”

One of my main problems is that in general I find the writing to be very lazy. Many sentences are ambiguous, too colloquial, poorly worded, and/or could be interpreted as incorrect. For example, the second sentence of the introduction seems to imply that measure qubits are unphysical, whereas clearly they are physical qubits too. The first sentence is also very strange. Stabilizer codes are just subspaces of a larger Hilbert space. I don't think an 'error-correction scheme' is a well-defined technical term, but to me it implies, among other things, hardware with stabilizer readout circuits and a decoder that carry out the dynamics of quantum error correction. Therefore, the opening statement 'Many quantum error correction schemes can be classified as stabilizer codes,...' does not make very much sense to me. While the introduction seems to be particularly bad, sentences like these continue throughout the manuscript. Given the number of coauthors, I believe that this work could have been written up with much more care. I am unwilling to recommend publication unless some effort is made to improve the writingsubstantially.

To address the referee's specific concerns, we have reworded the first two sentences of the introduction as follows:

Many QEC architectures are built on *stabilizer codes*, where logical qubits are encoded in the joint state of multiple physical qubits, which we refer to as *data qubits*. Additional physical qubits known as *measure qubits* are interlaced with the data qubits, and are used to periodically measure the parity of chosen data qubit combinations.

Additionally, we have made dozens of edits to increase the clarity and specificity of the paper. We highlight some of the more significant changes here:

- “However, achieving exponential error suppression in large systems is not a given, because typical error models for QEC do not include effects such as crosstalk errors.” now reads “**However, these results cannot be extrapolated to exponential error suppression in large systems unless non-idealities such as crosstalk are well-understood.**”
- Code distance is now “**defined as the minimum number of physical errors required to generate a logical error.**”
- The paragraph starting with “In this work, we run two stabilizer codes.” now includes formulas relating number of qubits to code distance for both the repetition code and surface code
- “typical coherence time” now reads “**mean coherence time**”
- The following sentence describing the reset gate is now more specific: “First, we use the reset protocol introduced in Ref.[XX], which removes population from excited states (including non-computational states) by **sweeping the frequency of**

each measure qubit through that of its readout resonator.”

- The paragraph introducing detection event fraction and the differences in the time boundary rounds has been reworked for clarity
- “Previous experiments had observed rising detection event fractions with **number of rounds**”
- “The exception is an error during the CZ gates, which may cause detection events offset by 1 unit in time and space” now reads “**The exception is an error between the two CZ gates in each round, which produces detection events offset by 1 unit in time and space**”
- The description of the depolarizing model simulations has been rewritten for clarity.
- “The relationship between $1/\Lambda$ and the component error rates is roughly linear” now reads “**The relationship between $1/\Lambda$ and each of the error probabilities is approximately linear**”
- “Logical errors can still occur after post-selection, for example with two simultaneous errors.” now reads “**Logical errors can still occur after post-selection if two or more physical errors flip the logical state without generating a detection event.**”
- “we show that a system with 21 superconducting qubits is stable” now reads “**we demonstrate stable error detection event fractions while executing 50 rounds of stabilizer measurements on a Sycamore device.**”

In the abstract, I don't agree with the statement 'QEC also requires that the errors are local...'. One could come up with a set of non-local errors that are correctable with some code according to the Knill-Laflamme error correction conditions. I understand that local errors are important for the protocol presented here, but I think this statement could be made more precise.

We agree, and have removed the statement that local errors are required for QEC in general from the abstract.

I think it is important that the original Dennis et al. paper is cited as an original work introducing the surface code, and for introducing the minimum-weight perfect matching decoder, the method of decoding used here. Likewise Kitaev's original 'Fault tolerant quantum computing with anyons' paper should be cited as the seminal work where the toric code is introduced.

We have added these citations to the main text.

In the caption for figure 4d it says the data shows good agreement with the model, although the data seems to diverge from the straight lines that are plotted. In the main text it is claimed that this is due to crosstalk and leakage, but perhaps this should be mentioned in the caption and/or marked on the figure as well. I see that more details are given in the supplemental material to explain this discrepancy, but is it possible to calculate how many errors are due to leakage and crosstalk to a leading order by comparing the model with a straight line fitted to the data?

The relevant caption now mentions “**Depolarizing model simulations which do not include leakage or crosstalk...**”. Additionally, we have added a table with the linear fits for experiment and simulation to the Supplement, added a discussion of additional sources of unmodeled error that are present in the surface code but not in the repetition code, and added more specificity in the main text in our description of the $d=2$ results.

You should add references [Harper and Flammia, PRL 122, 080504 (2019)] and [Willsch et al. Phys. Rev. A 98, 052348 (2018)] to your list of references in the final table in your supplemental material, and perhaps the work of [Heeres et al. Nat. Commun. 8, 94 (2017)] could be included too.

Thank you for the suggestions, all of which have been added to the table.

Reviewer Reports on the First Revision:

Referees' comments:

Referee #1 (Remarks to the Author):

Dear authors,

thank you very much for your answers.

All my points have been properly addressed, but I still have a comment on the circuit shown in Figure S10.

The authors mentioned (in their answer to my previous question on this figure) that the circuit as implemented was convenient for software scheduling purposes. What does ‘convenient’ exactly mean? When scheduling the gates, only the gate dependencies were considered or there are other constraints coming from for instance classical electronics or the quantum device itself that had to be taken into account? I would appreciate if the authors could elaborate a bit more on this aspect (scheduling of the gates) in the manuscript and how that can affect the performance of the circuit.

Referee #2 (Remarks to the Author):

The authors have taken the points raised by the referee's into account. With the modifications I would recommend the manuscript for publication in Nature.

Referee #3 (Remarks to the Author):

I believe the revisions the authors have made have improved the quality of the manuscript. I am happy to recommend publication.

It has been my pleasure to referee this work.

Benjamin J. Brown

Author Rebuttals to First Revision:

We thank all of the reviewers for their diligent reviews. To Reviewer 1, we again thank them for their detailed interest in our experiment. We have added the following text to Supplementary Section VI.

Note that the placement of some of the Hadamard gates is flexible. For example, for the top measure qubit shown in Fig.~\ref{fig:d2-2}, the Hadamard could in principle be placed anywhere prior to the first CZ which is operating on that qubit with no change to the stabilizer. We first rule out placing the Hadamard \textit{during} the CZ gate, as the effect of simultaneously operating single- and two-qubit gates on gate fidelities has not been explored in depth yet. With two possible locations remaining, our software defaulted to placing that Hadamard as early as possible. However, a potentially better choice would have been to defer the Hadamard until immediately before the CZ gate, so that the measure qubit would stay in $|0\rangle$ for longer and be exposed to less decoherence. Since the length of the CZ gate is only 26 ns compared to a mean T_2 of $19\ \mu\text{s}$, we expect this oversight to have a negligible effect on the overall results.

Finally, we note one minor technical correction that was uncovered when writing the Methods section - the number of experimental shots acquired and analyzed in the repetition code data was 80,000, not 76,000.